# The Current States, Challenges, Ongoing Efforts, and Future Perspectives of Pharmaceutical Excipients in Pediatric Patients in Each Country and Region

**DOI:** 10.3390/children9040453

**Published:** 2022-03-23

**Authors:** Jumpei Saito, Anjali Agrawal, Vandana Patravale, Anjali Pandya, Samuel Orubu, Min Zhao, Gavin P. Andrews, Caroline Petit-Turcotte, Hannah Landry, Alysha Croker, Hidefumi Nakamura, Akimasa Yamatani, Smita Salunke

**Affiliations:** 1Department of Pharmacy, National Center for Child Health and Development, Okura 2-10-1, Setagaya-ku, Tokyo 157-8535, Japan; yamatani-a@ncchd.go.jp; 2Drug Product Development, Bristol Myers Squibb, 181 Passaic Avenue, Summit, NJ 07901, USA; anjali.agrawal@bms.com; 3Department of Pharmaceutical Sciences and Technology, Institute of Chemical Technology, Mumbai 400019, India; vb.patravale@ictmumbai.edu.in (V.P.); ak.pandya@ictmumbai.edu.in (A.P.); 4Department of Biomedical Engineering, Boston University, 44 Cummington Mall, Boston, MA 02215, USA; sforubu@bu.edu; 5Department of Pharmaceutics and Pharmaceutical Technology, Niger Delta University, Amassama 560103, Nigeria; 6Medical Biology Centre, School of Pharmacy, China Medical University-Queen’s University Belfast Joint College (CQC), Queen’s University Belfast, 97 Lisburn Road, Belfast BT9 7BL, UK; m.zhao@qub.ac.uk (M.Z.); g.andrews@qub.ac.uk (G.P.A.); 7Therapeutic Products Directorate, Health Canada, Government of Canada, Ottawa, ON K1A 0K9, Canada; caroline.petit-turcotte@hc-sc.gc.ca; 8Office of Pediatrics and Patient Involvement, Health Canada, Government of Canada, Ottawa, ON K1A 0K9, Canada; hannah.landry@hc-sc.gc.ca (H.L.); alysha.croker@hc-sc.gc.ca (A.C.); 9Department of Research and Development Supervision, National Center for Child Health and Development, Tokyo 157-8535, Japan; nakamura-hd@ncchd.go.jp; 10UCL School of Pharmacy, 29-39 Brunswick Square, London WC1N 1AX, UK; s.salunke@ucl.au.uk

**Keywords:** excipients, pediatric patients, age-appropriate dosage form

## Abstract

A major hurdle in pediatric formulation development is the lack of safety and toxicity data on some of the commonly used excipients. While the maximum oral safe dose for several kinds of excipients is known in the adult population, the doses in pediatric patients, including preterm neonates, are not established yet due to the lack of evidence-based data. This paper consists of four parts: (1) country-specific perspectives in different parts of the world (current state, challenges in excipients, and ongoing efforts) for ensuring the use of safe excipients, (2) comparing and contrasting the country-specific perspectives, (3) past and ongoing collaborative efforts, and (4) future perspectives on excipients for pediatric formulation. The regulatory process for pharmaceutical excipients has been developed. However, there are gaps between each region where a lack of information and an insufficient regulation process was found. Ongoing efforts include raising issues on excipient exposure, building a region-specific database, and improving excipient regulation; however, there is a lack of evidence-based information on safety for the pediatric population. More progress on clear safety limits, quantitative information on excipients of concern in the pediatric population, and international harmonization of excipients’ regulatory processes for the pediatric population are required.

## 1. Introduction

Pediatric patients have different requirements compared to adults, regarding pharmacotherapy [1]. Flexible dosing, appropriate excipients, ease of administration, and dosage form acceptability or palatability are some of the key parameters for developing formulations appropriate for different age groups of the pediatric subset [2]. Pharmaceutical excipients are no longer considered inert in general, as new evidence suggests that there may be safety concerns with some excipients when used in products for the pediatric population, especially with younger age groups [3]. For example, immaturity of the metabolic and clearance functions, in neonates and infants, can lead to the toxicity of excipients such as propylene glycol [4,5], benzoic acid, and benzoates [6]. These excipients should, hence, be used with caution in noticeably young patients, such as preterm neonates. Paraben-containing drugs, injectable saline, and water for injections should be contraindicated in jaundiced newborn infants when the high-affinity albumin-binding sites approach saturation [7,8]. The use of benzalkonium chloride in the pediatric population has been reported to cause dose-related bronchoconstriction, especially in pediatrics who have asthmatic conditions, and has been related to the precipitation of respiratory arrest [9]. Ethanol, which is used as a solvent or a preservative agent in oral liquid preparations, has severe acute and chronic adverse effects in the pediatric population [10]. Flavoring agents may be used to impart taste, improve palatability, and thus, improve medication adherence. They are used in comparatively small amounts so that exposure is relatively low. However, there are safety concerns associated with flavoring substances, with respect to the potential risk of genotoxicity, allergy, and sensitization [11]. There are safety and biopharmaceutical challenges, of commonly-found excipients, in pediatric formulations [12,13,14].

Many excipients have no available safety data to justify their use during regulatory approval of the pediatric drug products. While the maximum oral safe dose for several kinds of excipients is known in the adult population [15,16], the acceptable excipient levels in pediatric patients (including preterm neonates) have not been established yet, due to the lack of evidence-based data [5,6]. Furthermore, the guidance or recommendation on excipient use for the pediatric population varies between countries around the world.

This article summarizes country-specific perspectives, including: (1) the current state on the safety assessment of pharmaceutical excipients, in formulations for both adults and pediatrics (including the disclosure status of the excipients in the prescribed drugs) and challenges in excipient regulation; (2) ongoing efforts for ensuring the safety of excipients, for the pediatric population, through the pediatric drug development in Africa, Australia, Canada, China, Europe, Japan, and the United State of America (US). Additionally, country-specific perspectives were compared, and aspects of past and ongoing collaborative efforts on excipients used for the pediatric population are presented (Figure 1).

## 2. Country-Specific Perspectives

### 2.1. Africa

#### 2.1.1. Current State and Challenges in Excipients Regulation

Africa is represented by Nigeria and South Africa.

Nigeria, through the National Agency for Food and Drug Administration and Control (NAFDAC), codifies regulations on pharmaceutical excipients in pediatric patients in several documents. These are: the Guidelines for Registration of Pharmaceutical Products (GRPP), Clinical Trials in Paediatric Populations (CTPP), Summary of Product Characteristics (SmPC), Product Information Leaflet (PIL), Label Guidance, and Non-nutritive Sweeteners Prohibition in Drug Regulation.

The GRPP, a general guideline for all pharmaceutical products, includes a consideration of excipients for children, referring to the World Health Organization’s (WHO’s) “Guidelines on development of pediatric medicines: points to consider in the choice of excipients” document. It also specifically recommends the use of additional guidelines for excipients to be avoided, including azo dyes. For information on acceptable (established or non-novel) excipients and their concentrations, the GRPP refers to other resources, such as the US Food and Drug Administration (FDA) inactive ingredient guide list and the Handbook of Pharmaceutical Excipients. Excipients should not be used in concentrations outside of established range. Ranges outside the established concentrations are not accepted unless the data is supported by the appropriate process. Furthermore, other available guidelines should be referred, which discuss particular excipients to avoid [17]. Additionally, it only allows colors and flavors permitted by other competent authorities, such as the FDA and the European Medicines Agency (EMA) [17]. Products for registration are required to comply with the label, SmPC, and PIL specifications in the relevant guidelines. These mandate a listing of excipients, but they do not require quantifications [18]. The CTPP guidelines recognize the toxicity of some excipients in children, depending on their ages, and the need for substitutions in clinical trials [19].

For novel excipients, these are not accepted by NAFDAC. A novel excipient is defined as “one that has not been used (at a similar level and by the same route of administration) in a product approved by a Stringent Regulatory Authority or by the WHO”.

Apart from a reference to product registration by a competent health authority, and the need for the label to have “adequate warnings where necessary”, the Guidelines for Registration of Imported Drug Products in Nigeria (Human and Veterinary Drugs) contains no specific reference either to children or to excipients [20]. The Non-nutritive Sweeteners in Drug Products (Prohibition) Regulations 2019 permits only four sweeteners, as well as specifies the maximum contents for them: acesulfame potassium, aspartame, neotame, and saccharin up to, respectively, 15 mg, 40 mg, 2 mg, and 2.5 mg/kg body weight [21]. Any product containing any other sweeteners as excipients, apart from these, would be regarded as “adulterated” and hazardous to health. However, the Acceptable Daily Intakes (ADIs) for these four excipients are for adults and not necessarily children, and the regulation made no specific mention of children. 

There have been several reports on excipient toxicity in Nigeria. The “My Pikin” tragedy, in which 109 children died from acute kidney disease caused by the substitution of the (non-permitted) toxic diethylene glycol for the excipient propylene glycol, used as a solubilizer for paracetamol, is the most recent [22]. It should be noted that this unscrupulous use of diethylene glycol, as a cheap adulterant alternative to propylene glycol, has also led to deaths in several other countries, including the USA, Bangladesh, and India. This is one case of an unlicensed pharmaceutical excipient of concern in children. With licensed products, a survey by Soremekun et al., 2019, to assess the presence of excipients of concern in children, found alarmingly high proportions of ethanol in oral liquid formulations commercially available in Benin City [23]. 

In the case of South Africa, the South African Health Products Regulatory Authority (SAHPRA) is the competent authority responsible for medicine regulation. In a guideline for the registration of human medicines, –Guidelines for Professional Information for Human Medicines (categories A and D)(SAHPRA) [24] under Section 2, addressing composition mandates that “full details of the qualitative and quantitative composition in terms of the active substance(s), and excipients where knowledge of which is essential for the proper administration of the medicine, should be provided”. Section 2 also requires the sugar status (for example, sugar-free) and quantity to be stated; where applicable, the same should be provided for other sweeteners. Compositional requirements extend to complementary (plant-derived) medicines containing alcohol, for which it states that “information about the ethanol content in the medicine should be included following the European Commission (EC)’s Guideline on Excipients in the Labelling and Package Leaflet of Medicines for Human Use” [24].

Overall, while a list should be given of all excipients in the product, expressed qualitatively only, the quantities of excipients with known pharmacological, as well as known or recognized action or effect–in addition to the sugars and sweeteners–need to be declared. This regulation covers flavors, pH adjusters, and others, such as printing inks. pH adjusters should be listed even if they are not present in the final product.

South Africa adopts the principle of reliance [25]. Thus, in general, this guideline, with certain exemptions, relies largely on the EC’s Guideline, as well as other stringent authorities or compendia, including the International Conference on Harmonization (ICH), in specifications related to pharmaceutical excipients in products intended for use in children. For example, preservatives and alcohol content above 2% need to be declared in the Patient Information Leaflet, as stated in the Guideline for Patient Information Leaflet for Human Medicines (categories A and D) [26]. This guideline also includes the need to specify all excipients known to have adverse effects in all patient groups, including children. Other relevant guidelines in South Africa include: the Quality and Bioequivalence Guidelines, with excipient labelling requirements based on the EC’s guideline [27], the Variations Addendum for Human and Veterinary Medicines, which mandates justification, based on approvals by recognized regulatory agencies, in changes to specifications or limits for excipients, [28] and where the safety and efficacy of these are shown by bioequivalence studies. 

#### 2.1.2. Ongoing Efforts

While there is some attention given to excipients, both in general and specific to children [23,29], these reports and surveys highlight the need for a more robust regulatory focus on excipients in children’s medicines in Nigeria and South Africa. Working toward both improvements of safe excipients use and appropriate drug development for the pediatric population are required. In Africa, almost every country has a national medicines regulatory authority, but its functions and expertise are varied among countries [30]. Although they have no specific pediatric regulations, excipient regulation in Nigeria is, at least, followed by the EC, FDA, and ICH guidelines.

### 2.2. Australia

#### 2.2.1. Current State and Challenges in Excipients Regulation

If a drug contains an excipient that is being used for the first time in a therapeutic product in Australia, the nonclinical overview must include an assessment of the safety information. Nonclinical data must be provided for a new excipient, an excipient with a new route of administration, or an excipient with an increased daily dose; otherwise, the application must include an explanation for not giving data. Submitting evidence of Good Manufacturing Practice to the authority for the manufacture of higher-risk excipients, such as those of human or animal origin, is also required [31]. In the case of excipients with coloring characteristics, natural coloring, or the excipients previously used for the prescription medicines can be used without further evaluation of toxicology data. For example, if sponsors wish to use a fruit extract, such as Vaccinium myrtillus (bilberry), as a natural coloring in medicines, no additional toxicological data is required. The TGA indicates the list of the coloring agents that can be used in medicines for topical and oral use, as well as those that do not require evaluation of toxicology data [32]. If the used colorings in topical or orally-administered medicines are not in the published list, it is necessary to provide the data about the chemical name and its properties, along with the clinical and toxicological tests. For colors included in the EU regulations for food additives [33], the evidence of compliance with the directive is required. For other new colorings, quality data that are consistent with the EU Guideline on excipients, in the dossier for the application for marketing authorization of a medicinal product, are required [34]. This guideline includes some specific considerations for pediatric populations. For example, coloring agents with documented safety risks, e.g., azo dyes and other synthetic coloring agents, should not be used in medicinal products for pediatric use when only intended for aesthetic purposes. If there are any, including reports of all human clinical studies of the coloring are required. Regarding the product labeling, there is no requirement to declare all excipients on medicine labels. Only those excipients specified in Therapeutic Good Order (TGO) No. 79 are required to be declared in the ingredients section of the panel. Reference to color, fragrance, or flavor (e.g., red capsule, strawberry flavor) is generally considered to be acceptable without justification [35].

#### 2.2.2. Ongoing Efforts

In 2019, The TGA is seeking opinion on whether excipient ingredients should make accessible on the Australian Register of Therapeutic Goods (ARTG) [36]. When implemented, excipient ingredient names is published in ARTG public summaries for prescription medicines, biologicals (cell and tissue therapies), non-prescription medicines (registered and listed, including complementary medicines), and other therapeutic goods.

Excipient component names are now only viewable in the TGA’s internal view of the ARTG. Because consumers are increasingly turning to internet sources for health information,, the TGA considers that publishing excipient ingredients would assist the safe use of therapeutic goods. This effort is only getting started., so it is unknown if this effort contributes to ensuring the safe use of pharmaceutical products for pediatric patients.

### 2.3. Canada

#### 2.3.1. Current State and Challenges in Excipients Regulation

In Canada, the drug development process is bound by the Canadian Food and Drugs Act (F&D Act) and the Food and Drugs Regulations (F&D Regulations). Within the F&D Act, the F&D Regulations does not define excipients, but rather, at C.01.001, it defines non-medicinal ingredients as ‘*a substance- other than the pharmacologically active drug- added during the manufacturing process and that is present in the finished drug product*’ [37]. As a regulatory requirement to the Act and Regulations, all non-medicinal ingredients (which, for the purpose of consistency with other authors, will be hereinafter called “excipients”) are listed in the Canadian Product Monograph or Prescribing Information. Health Canada has published guidance documents referencing the appropriate use of excipients, such as *Quality (Chemistry and Manufacturing) Guidance: New Drug Submissions (NDSs) and Abbreviated New Drug Submission (ANDSs)* (see Section P.4 in reference [38]) and *Labelling of Pharmaceutical Drugs for Human Use* [39]. Canada has also adopted ICH quality and safety guidelines and, as such, adheres to the recommendations found therein [40]. It should be noted that the recommendations made by the various sources of guidance, regarding excipients, are highly encouraged and voluntary, at the moment, until such time as the F&D Act and F&D Regulations provide clear, direct authority, with respect to excipients. 

Excipients are an integral aspect of the formulation, which influences the drug product’s stability and delivery, but they may also have an impact on its safety profile. Typically, a justification for the excipients used in the formulation must be adequately addressed in the drug submission package. This includes flavorants, which can be proprietary mixtures, but their components should be disclosed in the submission package for review and qualification, if applicable, as per F&D Regulation C.08.002(2)(c). Supplementary information will be required in the submission in the case of novel excipients (i.e., those which have never been used in an approved product in Canada or are being proposed in a new context of use, such as a new route of administration) as well as when intended for use in vulnerable populations, such as pediatrics [38].

The pediatric population poses an additional challenge when selecting excipients in the design of formulations intended for these patients. When developing liquids, granules, or micro-dose formulations, for example, in addition to considerations for stability, solubility, and release, there are also important aspects related to flavor and palatability that must be considered, especially for populations that are 8 years and younger, as they directly impact compliance [41,42]. 

Considerably less attention has been paid to excipient use in the pediatric population, and as such, there are significant gaps in understanding the safety and efficacy in this population [43]. Current knowledge concerning the safety and quality of excipients stems, predominantly, from previously-collected adult and pediatric data [14,44]. In some circumstances, extrapolation of safety and efficacy of excipients from adult data may not be sufficient due to the heterogeneity of the pediatric population and, specifically, of the pharmaco/toxico-kinetic properties of the excipients for neonates and infants [45,46].

Drug compounding is a common practice in Canadian pharmacies and hospitals to assemble tailored medicines, or combinations of medicines, for pediatric patients. In many instances, compounding medicines for individual patients is done effectively and accurately in accredited compounding pharmacies. While drug compounding is not within Health Canada’s mandate to authorize [47,48], it is carefully regulated by Canada’s various Colleges of Pharmacy (at the provincial and territorial level), which outline the specific documentation, training, and equipment required, as well as the circumstances under which products may be compounded in Canada. Poor compounding practices can result in serious quality issues, such as changes in the ratio of the active ingredient to excipient(s), which may result in changes to pharmacokinetics, exposure of the active ingredient, and toxicity. This may impact the safety and/or effectiveness of the drug product. A federal framework to regulate commercial compounding and its oversight is currently being developed in Canada [49]. 

In addition to the implicit issue of heterogeneity among pediatric patients, the choice of excipients is further complicated by multiple additional factors, including disease-specific factors, physiological limitations (e.g., solubility, requirements around the route of administration, etc.), and hypersensitivities, and/or rare adverse reactions to excipients. Some excipients have been previously identified as toxic in pediatric populations and should be avoided (e.g., benzyl alcohol for neonates; polysorbate 80 for infants under 1 year old, etc.) [12,50]. Some excipients should be used with caution in certain age groups and/or with a daily dose limit (e.g., methylparaben for infants under 2 months of age; ethanol for children under 6 years, with a maximum amount of 5% *vol*/*vol*, etc.) [51]. Furthermore, seemingly innocuous excipients related to palatability, such as sweeteners (natural or artificial), flavoring agents, and color, can lead to a number of safety issues [14,42]. Sweeteners (such as sucrose, aspartame, or saccharin) should be avoided, as they contribute to tooth decay, changes in bioavailability of the drug product (or concomitant medications), impact blood glucose levels, or hold a carcinogenic potential (e.g., saccharin in children under 3 years of age). Flavorants (for example, peppermint oil) have been associated with some toxicity, by inhalation and ingestion [14], and should not be used in younger children. Dyes (e.g., azo-dyes, quinolones, etc.) are useful to facilitate recognition and even appeal of drug formulations; however, most should be avoided in children, as they present risks of hypersensitivity reactions and cross-reactivity with other active ingredients [12,52]. As for all drug products marketed in Canada, pediatric drugs are subject to post-market monitoring by Health Canada, including the evaluation of adverse reaction reports [53].

Quality evaluations of excipients used in drug formulations, conducted by Health Canada, are based on the Health Canada Guidance Document *Quality (Chemistry and Manufacturing): New Drugs Submissions (NDSs) and Abbreviated New Drugs Submissions (ANDSs)*, as well as ICH guidelines, for impurity qualification (e.g., ICH Q3, M7). Additional data supporting safety assessments (nonclinical and/or clinical) may be requested, when needed, on a case-by-case basis (see below), including in the case of excipients deemed to be novel for pediatrics [38].

#### 2.3.2. Ongoing Efforts

During the quality assessment of most excipients, Health Canada refers to the specifications (chemical, analytical) outlined in the various official pharmacopeial monographs, such as the United States Pharmacopeia (USP) or Ph.Eur. (when available). The European Paediatric Formulation Initiative (EuPFI) Safety and Toxicity of Excipients for Paediatrics (STEP) database for pediatric excipients [50,54,55] is a useful, supportive source of information, but it is not recognized as an official pharmacopeia in Canada. For novel excipients not supported by pharmacopeial monographs and/or aforementioned excipient databases, additional data to support the safety (including toxicology studies) and quality assessments may be required as part of the drug submission. 

Regarding the safety evaluation, there are several physicochemical properties of excipients (e.g., particle size, polymer length, relative amount to the active pharmaceutical ingredients (APIs), etc.) that can impact the pharmacokinetic properties (e.g., solubility and bioavailability) and, potentially, the safety profiles of APIs. New drug candidates increasingly include new APIs with low solubility, driving the need for novel excipients to improve solubility and bioavailability [56,57], in addition to providing stability, palatability, and acceptability. These excipients are also essential to the development of new formulations, including some that are tailored for pediatric formulations (e.g., micro-doses, granules, pre-filled syringes, etc.). There is an expectation that these novel, fit-for-purpose excipients should be characterized to the same extent as a new API in terms of quality and safety (nonclinical and clinical) [58]; however, there is currently no designated pathway for the regulatory approval of novel excipients as stand-alone new chemical entities. Their qualification, in combination with a new API, increases the risk for sponsors (as any issues with the excipient’s application would jeopardize the approval of the drug candidate). The excipient-related risk may also lead sponsors to refrain from moving promising candidates forward in development [57], especially in the case of pediatric formulations [44], where the return on investment is likely lower.

Impurities and degradation products can originate from excipients. While some excipient-related impurities and degradation products are well characterized in terms of the pharmacokinetics of excipients, this is often not the case for those originating from novel excipients. There is currently no published guidance in Canada to qualify the safety of impurities originating from excipients. Instead, Health Canada’s approach has been to follow a weight of evidence rationale and to evaluate the safety and qualification of excipient-related impurities, including degradation products, as is done for those originating from the API. This approach, however, does not consider any pharmacokinetic and/or physiological differences between adults and the different pediatric age groups. 

The general principle for safety evaluation, regarding excipients in pediatric formulations, is described in ICH E11 (R1) and S11, which reflect the general approach at Health Canada. The weight of evidence is based on available information, and the requirements for a juvenile animal study for testing excipients are stated in ICH S11 [59].

The absence of harmonized standards or requirements for assessing the safety of novel excipients, both in general and in the context of developing pediatric formulations, can lead to different expectations for sponsors and regulators. This lack of harmonization and clarity around regulatory requirements may result in delays for both drug development programs and market authorizations, due to uncertainties around testing novel excipients in pediatric formulations.

### 2.4. China

#### 2.4.1. Current State and Challenges in Excipients Regulation

China is currently one of the most promising pharmaceutical markets, and new regulatory requirements are leading to important changes, especially regarding excipients used in drug manufacturing. Article 11 of the Pharmaceutical Administration Law, promulgated in 2001, stipulated that those excipients used for pharmaceutical production should meet the requirements for medicinal use [60]. However, this general statement led to quite inconsistent approaches. Some Chinese provinces regulated excipients as APIs, while others did not regulate them at all. In 2005, the China Food and Drug Administration (CFDA) issued the Pharma Excipient Dossier Requirements for industry, proposing excipient registration according to the same process as APIs, with a stand-alone review by the Centre of Drug Evaluation (CDE) for import and novel excipients, and a review by the local authorities for excipients described in the Chinese Pharmacopoeia (ChP). The co-review process for excipients went into law in December 2017. According to CFDA announcement No. 146, all pharmaceutical excipient manufacturers or owners, domestic or foreign, must submit their dossiers to the CDE. In 2021, the new CFDA regulations, “Announcement of the CFDA on Adjusting Matters Concerning the Review and Approval of Active Pharmaceutical Ingredients, Pharmaceutical Excipients and Pharmaceutical Packaging Materials (No. 146, 2021)” decided that the approval of pharmaceutical excipients is conducted as a part of a drug product application. Relevant pharmaceutical excipient manufacturers need to submit pharmaceutical excipients-filing dossiers (i.e., The Drug Master File (DMF)) to the CFDA. After the drug application passes CFDA approvals, the pharmaceutical excipients will automatically pass the CFDA approval. Pharmaceutical excipient manufacturers need to submit annual product quality management reports to the CFDA to keep their DMF registration number active after obtaining the CFDA approval.

The classification of excipients, as listed in the ChP, was proposed in 2018 [61], which has, since then, served as the basis for the dossier requirements for excipients. The level of detail required for the registration dossier is based on the excipient classification, as revised by the NMPA in July 2019 [62] Excipients are classified into products with or without a history of use in approved drugs. The latter includes completely new molecules, as well as molecules with simple changes to their structure or a changed route of administration. Products with a history of use are, in turn, divided into two groups: excipients that are included, or not included, in the ChP or the pharmacopoeias of the European Union (EU), the US, United Kingdom (UK), or Japan. The current ChP edition includes 270 excipient monographs. However, many drug manufacturers cannot yet register their products in China, and the ChP has not yet been fully harmonized with other international compendia, which forces global pharmaceutical companies to perform additional comparisons of methods and cross-validation checks. The goal for the 2020 edition is to add another 100 excipient monographs and promote the harmonization with other national pharmacopeias in the EU, the US, and Japan. 

As confirmed in announcement No. 56, which went into effect on 15 August 2019, the NMPA has also identified low-risk excipients that are exempt from mandatory registration. Apart from corrigents (taste masking agents), flavors, spices, pH adjusters, and inorganic salts, the exemption list for low-risk excipients also includes pigments (colorants) and inks—more precisely, iron oxide, plant carbon black, and cochineal, as well as benzene-free inks for capsule inscription. Generally, the exempt excipients can be used without going through the complex review and registration process. However, the final registration exemption of an excipient needs to be confirmed by the CDE and depends on how the excipient is used in the drug formulation.

Children of different ages, ranging from newborn to 17 or 18 (some are defined as 14), have great physiological differences. The tolerance of children to excipients is different from that of adults. Children in different ages also have different tolerances to the same excipient. When developing children’s preparations, it is necessary to strictly consider the age range, analyze the tolerance of children to excipients in the particular age group, and avoid adverse reactions and side effects caused by improper selection of excipients or excessive use of them. The Guiding Principles for Pharmaceutical Development of Paediatric Medicines (Chemicals) (for Trial Implementation), issued by the Drug Evaluation Center of CFDA on 31 December 2020, clearly states: “The selection of suitable excipients is one of the key factors in the pharmaceutical development of pediatric medicines. When choosing excipients, children’s age, weight, degree of physical growth, administration frequency, proposed course of treatment, commonly used combination drugs etc. should be fully considered. The least variety and number of excipients should be used as far as possible while reducing risk and ensuring efficacy, stability, palatability, microbiological control and dose uniformity of the product”.

In the pharmaceutical industry, the selection of excipient type and the determination of the appropriate amount are very important in the development of new products. In terms of generic drug products, the excipients used in the reference product can be considered as a solid base for determining the type of excipients in the product under development, and the amount of excipients can be optimized according to the reverse engineering analysis results of the reference drug product. As for innovative pharmaceutical products, there is no reference product available. Therefore, the type and number of excipients can only be selected on the basis of the nature of the drug substance, the intended dosage and dosage form, as well as the use of excipients in other products on the market, through scientific screening and risk assessment. Excipient selection is the most challenging in the case of Class 2 modified drugs, especially Class 2.2 modified formulations, because these formulations can be commercialized through bio-equivalence trials; hence, it is difficult to prove the safety of excipients through pre-clinical studies and clinical pharmacological, as well as toxicological, tests. Due to the lack of authoritative public data to support, there is a great challenge to the rationality of excipient selection for Class 2 modified new drugs. 

#### 2.4.2. Ongoing Efforts

The Chinese regulatory authorities also engaged in an intensive exchange with the US FDA, the respective EU authorities, and the European Directorate for the Quality of Medicines and Health Care (EDQM). Moreover, China has been a member of the ICH since 2017. The implementation of the ICH guidelines in China is an ongoing process.

The determination of excipient type and amount has been a consistent challenge for the researchers of generic drugs and Class 2 improved new drugs. With the continuous development of the pharmaceutical industry, it is believed that there will be more public information available as reference in the future. For instance, the FDA Inactive Ingredients Database was revised in October 2020, where the maximum daily exposure of excipients has been added based on the amount of excipients used in the original unit dose formulation, which greatly broadens the reference range for the amount of excipients used in generic and modified drug product development.

### 2.5. Europe

#### 2.5.1. Current State and Challenges in Excipients Regulation

A guideline on ‘Excipients in the labeling and package leaflet of medicinal products for human use’ released by the EC. Excipients with a known action or effect are listed in the amended guideline. and, therefore, must appear on the labeling of all medicines in the EU [63]. The Annex also contains the safety information for the specified excipients that must be included in the medicine’s package leaflet. The EMA’s website has background information on the safety of individual excipients. [64].

Regarding the disclosure of quantitative information on pharmaceutical excipients, Article 59(1)(f)(iv) requires the full qualitative composition (inactive substances and excipients) and the quantitative composition of active substances to be included in the package leaflet [65]. All excipient names on the labeling, package leaflet, and Summary of Product Characteristics (SmPC) must comply with the following. Individual excipients should not have proprietary names. Fragrance and flavor ingredients can be declared in general terms (e.g., ‘orange flavor’, ‘citrus perfume’); it is necessary to declare a recognized action or effect. For excipients that are categorized in a chemical group in the Annex but are not explicitly listed (e.g., other salts), the information applies unless justified. pH adjusters should be mentioned by name, and their function may also be indicated in the package insert, e.g., hydrochloric acid and sodium hydroxide for pH adjustment. For some of the excipients in the Annex, the information may be included in the warnings section of the package insert (i.e., pregnancy and lactation, pediatric use, undesirable effects, warnings and precautions, contra-indications). Additionally, it may be necessary to refer to the excipient warnings section from other sections in the package leaflet. In the case of ethanol, it will be necessary to refer to the excipient warnings section from those sections relating to effects on the ability to drive, pregnancy and lactation, information for children, etc.

A ‘threshold’ value is also included in the Annex. However the stated information is not a safety limit. Thresholds are expressed as the quantity of an excipient at the maximum daily dose (MDD) of the medicinal product, as indicated in the SmPC. When the text refers to the term ‘per dose’ it means the dose of the medicinal product. 

Regarding the toxicology study in juvenile animals, the aspects are the same as US regulations. When the use of an excipient, in drugs for the pediatric population, cannot be justified based on existing information sources, toxicology studies for an excipient in juvenile animals may be necessary [3].

#### 2.5.2. Ongoing Efforts

Regarding information on the safety of excipients, the EMA proposed safety limits for several excipients in the pediatric population, such as propylene glycol [5] and sorbitol [66]. In January 2014, the EMA proposed the inclusion of more detailed information on alcohol content in PILs, as well as alcohol content thresholds for different age groups, in a draft for the guideline on ‘Excipients in the label and package leaflet of medicinal products for human use’ [64,67]. Acceptable daily intake for artificial sweeteners, such as aspartame and saccharin, is also stated in the National Health Service United Kingdom (UK) [68,69]. However, this information is not specified for pharmaceutical excipients. Regarding the novel excipients, such as hydroxypropyl-β-cyclodextrin, there is insufficient safety data on pediatric patients, especially on neonates [70,71].

As an available database, the STEP database is a user-designed free resource that compiles the safety and toxicity information of excipients, which is manually extracted from selected information sources [50,54,55]. Currently, the database includes 75 excipients, most of which are used in oral dosage forms. O’Brien et al. conducted a pilot review, identifying excipients in parenteral products, used for pediatrics in India, from the STEP database [72], and found that, of the 30 identified excipients for 104 parental products that are commonly used in pediatric population in India, only 10 excipients were included in the STEP database. This study will also be extended to other countries, such as UK and Ireland, to identify the excipients used in parenteral products and prepare a comprehensive list of excipients used in parenteral products, as well as those to be included in the STEP database. Further efforts are required by the sponsors to share and declare non-confidential in-house data, on the STEP database, to be a useful database and prevent repeated studies on excipient safety.

### 2.6. India

#### 2.6.1. Current State and Challenges in Excipients Regulation

India has the largest adolescent (10 to 19 years) population in the world of around 253 million, with every fifth individual falling in this age group [73,74,75]. Another statistic depicted that, from 2009 to 2019, around one-fourth of the population in India fell in the age group of 0 to 14 years, whereas two-thirds were in the 15 to 64 years category [76]. Considering the developing nature of the Indian economy, the availability of pediatric formulations, especially in rural areas, is still a matter of concern. Along with the global pharmaceutical regulatory agencies, the Indian drug regulators have also acknowledged the need for regulation in the Indian pediatric scenario. The accessibility of information on pharmaceutical excipients for products is limited, except for drugs in the WHO’s model formulary for children, the WHO’s model list of essential medicines for children, and the National List of Essential Medicines. For the domestic approved medicine, the excipient name is not required to be specified, and only the information on the use (type) of each excipient (i.e., sweetener, flavoring agent, coloring agent, etc.) is available. The information about excipients (except preservatives, colors, and alcohol, depending on the content) is not written on the label, which could be a hurdle in case of any safety issues. The suitability of pharmaceutical dosage forms for pediatric use is restricted majorly because of the potency of active ingredients and the use of numerous excipients that form the bulk materials [13], thereby making compounding into a traditional method of pediatric formulation dispensing. Commonly used excipients in pediatric formulations include antimicrobial preservatives (benzyl alcohol, propylene glycol), sweeteners (sucrose, fructose, lactose), solvents (ethanol, propylene glycol), coloring agents (tartrazine, carmoisine), coating agents (methacrylic acid, methacrylate copolymers), flavoring agents, etc. [77].

The Indian constitution lists Drugs and Health, concurrently, as one of its most important segments that is governed by both Centre and State governments under the Drug and Cosmetics Act 1940 and Rules 1945. In the regulatory process, only excipients that are claimed or graded in the Indian Pharmacopoeia are controlled by the Food and Drug Administration in India. Central Drugs Standard Control Organization (CDSCO) mainly controls the approvals of drug products, including excipients, within the Indian Pharmacopoeia. The drug approval application requires the inclusion of the approximate composition of the drug product, including the quantity and the compendial status of each excipient used. However, concrete and specific guidelines for excipients to be used in pharmaceutical products are lacking. The Indian scenario for pediatric product regulation is different from that of other countries because the product development relies on clinical trial results, and protocols follow adult formulation research [78]. Indian clinical practice depends on safety and efficacy outcomes from pediatric research conducted in other countries. Lastly, the new draft of the innovation policy of the Indian government also does not have any section on pediatrics. However, Schedule Y of the Drugs and Cosmetics Act provides a special category of pediatrics in clinical testing that has to be followed [79]. Auditing and monitoring excipient manufacturing and supply chains are needed to ensure the production and distribution of optimum quality excipients. This has led to the development of the International Pharmaceutical Excipients Councils (IPECs), in different parts of the world, since 1991 [80]. The councils have been established with the core objective of developing regulations to induce improvement in the excipient quality and, ultimately, patient safety. Following the same objective, IPEC India was incorporated, in January 2014, as a non-profit organization, and later, it was also made a part of the IPEC Federation that collaborates with excipient industries. IPEC India provides a common domestic, as well as international, platform to the manufacturers, distributors, regulators, and end users, through several assessment procedures for pediatric excipients, including tiered toxicology testing [14].

#### 2.6.2. Ongoing Efforts

As a research study on excipients’ use in pharmaceutical products, Nasrollahi conducted a study on excipient exposure, among neonates, in a neonatal intensive care unit [74]. The total number of prescribed drugs was 5535, that composed 77 different drugs. The qualitative and quantitative information on excipients was available for only 35 and 15 drugs, respectively. Over 50% of the included 746 neonates were exposed to potentially harmful excipients. About 10% of neonates received sodium metabisulphite and sunset yellow FCF at a higher dose than acceptable daily intake. Furthermore, Ponceau 4R, which is concerning possible allergic reactions in the EU and must state a warning label, was used as a colorant. Due to its effect on neurocognitive development and behavior, Ponceau 4R is banned to use in some countries [74]. These excipients, in their adult dose, also have limited quantitative permissibility, which becomes even more stringent when developing a pediatric dosage form. Accidental overdose with any of these excipients can result in adverse toxicity conditions, and their suitability for pediatric use must be established. Special considerations for use of these excipients, based on factors like dose, frequency of administration, age, disease condition, and length of treatment, need to be thoroughly considered.

### 2.7. Japan

#### 2.7.1. Current State and Challenges in Excipients Regulation

In Japan, an evaluation on the importance of excipient use can be made by referring to the Japanese Pharmaceutical Excipients Dictionary (JPED), which is edited by the Japan Pharmaceutical Excipients Council in conjunction with the Ministry of Health, Labor, and Welfare (MHLW). It includes monographs from the Japanese Pharmacopoeia (JP) or Japanese Pharmaceutical Excipients (JPE). All non-monograph excipients that have been previously used are also included [81]. Each monograph lists the nonproprietary name and synonyms for the various routes of administration in approved drugs. It also includes the application and maximum dosages, The pharmaceutical product application is submitted to the regulatory authorities by the pharmaceutical company, typically containing all relevant details concerning the excipient. The data in each monograph contain only the safety data, including maximum dosage of pharmaceutical excipients for the adult population, and there is no evidence-based data for the pediatric population. Additionally, not all excipients reach these texts due to companies withholding data because of concerns about releasing proprietary information. The DMF system allows the manufacturers of new excipients to submit the detailed information (manufacturing methods, data, etc.) of excipients to the Review Authority (Japan’s Pharmaceuticals and Medical Devices Agency, PMDA). The registered manufacturing information is quoted as the necessary information for an approval review of the pharmaceutical products in which excipients are used. Japan’s DMF system permits API manufacturers to register their products directly with the PMDA. Hence, in Japan, when multiple drug manufacturers are required to use the same API for their finished drug products, the manufacturers do not need to turn over sensitive information in support of the finished drug registration. Instead, the PMDA uses the available information from the DMF to approve finished drug applications. DMF registration is voluntary in Japan. 

Regarding the accessibility of excipient information in the pharmaceutical products, the package inserts must list all ingredients used as excipients. When excipients listed in the JP, in the Minimum Requirements for Biological Products, or in the Radiopharmaceutical Standards, are used in products, the names and quantities of these excipients must be included in the relevant package inserts or on the containers or wrappers. If safety problems considered to be caused by excipients have appeared, the names and quantities of excipients must be included in the relevant package inserts or, if necessary, on the containers or wrappers of all prescription drugs [81]. However, this criterion is not always focused on the safety of pediatric patients. All ingredients, as a rule, except for the ingredients stipulated in the MHLW Notification [82], shall be included in the package insert because of the social responsibility to disclose as much information as possible related to drugs as life-related products. In accordance with “Guideline for Descriptions on Application Forms for Marketing Approval of Drugs, etc. under the Revised Pharmaceutical Affairs Law” [83], there is no obligation to indicate quantitative information on the contents of excipients included in the enteral product, except for the high-risk excipients such as ethanol, because the excipients in enteral products are considered ineffective after treatment, and these products were not developed with the intention of neonatal and/or pediatric administration. Furthermore, in enteral products, contained flavors are not specified (just indicated as “flavor”). In parenteral products, quantitative information on the contained excipients is sometimes available; however, some of the general excipients, such as pH buffering agents and isotonic agents, are not disclosed. Furthermore, there is no regulation on excipient use for pediatric patients in Japan. No evidence-based information on the safety of excipients and no beneficial guideline for excipient selection exist. In the development and regulatory process, the safety of excipients for the pediatric population, including neonates and infants, is not concerned. Preparing the excipient quantification lists, which have been used with the previous pediatric products or guidelines that aid the excipient selection for pediatric drug development, will be needed.

#### 2.7.2. Ongoing Efforts

At present, there is no new initiative, in regulatory aspects for pediatric excipients, in Japan. On the other hand, the safety assessment of excipient exposure to pediatric patients in clinical settings was attempted. Saito and their colleagues conducted a study on excipient exposure among neonates as the single-center observational study in a neonatal intensive care unit [84]. Furthermore, a nationwide multicenter study was also conducted to determine the status of excipients’ exposure to neonates [85]. In this previous study, quantitative evaluation, by using the information on excipients contents in the products, was conducted. However, most excipients in enteral products and some specific excipients, such as pH adjusters and flavoring agents, in parenteral products are not disclosed of their quantity, so accurate evaluation could not be done. No other attempt has existed. 

### 2.8. United States of America

#### 2.8.1. Current State and Challenges in Excipients Regulation

Excipient guidelines from the FDA is mostly based on IPEC recommendations. To ensuring the safety, guidelines addressed the safety tests generally required to determine the safety of a new excipient [86]. However, unlike drugs, testing for a new excipient should be evaluated on a case-by-case basis. The USP published the IPEC Safety Guidelines as the General Chapter on Excipient Biological Safety Evaluation Guidelines. The FDA guidance refers to the ICH Safety Testing Guidelines for conducting testing for new excipients. In the US, DMF systems exist for excipients to support medication applications. The IPEC-Americas Master File Guide is a format guide for DMF submissions that may be used to create uniform excipient information. [86]. Excipients, colorants, taste, essence, or material employed in their manufacturing are all classified as Type IV DMFs in the US. In support of a new drug application, DMFs can be utilized to give information to the FDA. [87]. Testing strategies for short-, intermediate-, and long-term usage are also discussed in the FDA regulation. The use of a “family approach” to assess the safety of related excipients, such as various viscosity or molecular weight grades of a polymer excipient, has recently been discussed between industry and the FDA. This technique is presently being discussed, and it is intended to allow some flexibility in the use of safety information that includes a variety of related excipients to support the safety of a specific class in the family when safety information unique to that class is not available. However, it is not clear whether this approach can apply to pharmaceutical products for the pediatric population. 

To obtain approval for pediatric products, juvenile toxicity studies must be conducted in representative animal species to demonstrate the safety of the drug and the excipients used in the drug. There is no separate approval process for excipients in pediatric products. Color additives and flavors, unlike other excipients, are regulated separately from therapeutic uses. These substances are evaluated for safety in processes outside of the drug review process. All color additives in the US are subject to premarket approval by the FDA. Color additives listed in 21 CFR, Parts 74 and 82, must be analyzed and batch-certified by the FDA. In the case of a new flavoring substance, such substances can be evaluated by the Flavor and Extract Manufacturers Association (FEMA) of the US Expert Panel to determine if they are Generally Recognized as Safe (GRAS). Flavoring agents are determined to be GRAS by the FEMA Expert Panel under the authority granted in Section 201(s) of the FD&C Act. To support the safe use of a particular flavor, references to the FEMA GRAS evaluations can be included in the product application. The restricted availability of and access to safety data, as well as uncertainty in extrapolating exposure and effects between adults and children or nonclinical animals and humans, complicate the safety qualification of excipients for pediatric usage. Although regulatory guidance provides some guidance on the safety assessment of excipients [86], there is a lack of uniformity on what is acceptable or essential to effectively assess the risk-benefit profile of an adjuvant in various pediatric demographics and disease states. When the use of an excipient in a pediatric medical product cannot be justified based on available information sources, toxicology studies in juvenile animals may be required. [3,86,88]; however, the standardized conduct of juvenile toxicology studies in a routine “box-ticking” manner is not considered appropriate. If the effects on growth and development have not been previously documented, the safety evaluation should focus on them. [88]. The juvenile toxicity study can be designed to assess the safety of both the excipient and the active substance [86,88,89]. Details of nonclinical juvenile toxicity studies, as well as any clinical safety evaluation undertaken by a pharmaceutical industry to certify excipients as part of a medicinal product, are not disclosed to the public.

The USP 35/National Formulary 30 lists over 40 different functional categories for excipients [90]. USP General Notices 5.20 and 5.60 require excipients (additives and processing aids) to be on labels and reported when used at levels >0.1% (based on the International Council for Harmonization Q3B). While IPEC-Americas considers that excipient and pharmaceutical companies should communicate openly regarding the potential for the presence of additives [91,92], this can include the use of confidentiality disclosure agreements during excipient/supplier qualification.

The quantities of the excipients included in the final product are not listed in the labeling of each product. Several excipients, such as alcohol and solubilizer, which may cause hypersensitivity or other adverse reactions, shall be included in the label along with the amount. If a drug includes one or more inactive substances that are linked to a major safety concern in pediatric patients (all pediatric patients, particular pediatric age groups, or subgroups), the risk must be disclosed on the label [93]. In general, a substantial safety risk associated with an inactive component should be detailed in the boxed warning, contraindications, and/or warnings and precautions section, as well as stated in the pediatric usage part.

The screening and careful selection of excipients in a pediatric medicinal product is, thus, challenging due to lack of appropriate guidance on safety qualification and risk assessment of excipients for pediatric formulations.

#### 2.8.2. Ongoing Efforts

The Inactive Ingredient Database (IID) is an open information database for pharmaceutical excipients that offers information on inactive substances (excipients) found in FDA-approved prescription formulations. This data can be utilized by the pharmaceutical industry to help create new drugs. Once an inactive component appears in an authorized drug product for a certain route of administration, it is no longer considered novel for new drug development reasons and may require a less thorough assessment. For example, if a certain inactive component has been authorized in a specific dosage form and potency, a sponsor could consider it safe for use in a similar way in a similar product. In this database, the maximum potency of each excipient per unit dose is available, including enteral formulation. If a new drug application intends to use an inactive ingredient at a level that exceeds any of the IID listings without reason, the FDA will reject it. An inactive ingredient is considered justified, for receipt purposes, if the proposed level is at or below the amount indicated in the IID for the corresponding route of administration of the drug product. The IID, on the other hand, does not yet give information on the various exposure models (e.g., Maximum Daily Intake (MDI) based on the labeled dosage guidelines)., safety in pediatric populations, and acute versus chronic use) that may be needed during such a technical review. Some values are difficult to verify in some circumstances since they are related with old products. In other cases, the use of the term “NA” in place of a maximum potency may be used when the quantity of the excipient is variable (e.g., pH adjusters that are indicated in the formula as “quantity sufficient”). The FDA has updated the IID database, allowing users to run electronic queries to acquire correct MDI and MDE information for any route of administration for which data is available. MDE is the total amount of the excipient that would be taken or used in a day, based on the MDD of the drug product in which it is used. MDE is calculated as the dosage unit level of the excipient multiplied by the maximum number of dosage units recommended per day. MDE may also be referred to as MDI for oral drug products. When determining excipient MDE, the FDA will evaluate the applicant’s rationale for an MDD if it is not included in the product labeling.

IID does not differentiate between adult and pediatric products currently. The maximum allowable dose of excipient listed in IID may not be safe for pediatric use if the excipients have potential for causing any harmful effects due to patient age, so additional studies or precedence of use, in the same age group with similar use duration, may be needed to justify use of such excipients in pediatric products.

A risk-benefit approach should be used in safety assessments, as opposed to an approach where the safety assessment is only in the context of potential risks. This is particularly important, as most pharmaceutical products could not be manufactured without the use of excipients. In situations where the benefit-risk cannot be adequately characterized on the basis of prior use in pediatric patients, use of the excipient cannot be bypassed, but the therapeutic benefit of the drug is sufficient, it may be useful to proceed carefully and assess safety in the clinical setting.

Efforts are needed from both pharmaceutical and excipient manufacturers to fill in the gaps and identify the best practices and types of data needed for the safety assessment of novel excipients. For instance, the Novel Excipients Working Group (members from IPEC-Americas and the IQ Consortium) and a similar group, formed within IPEC-Americas, are currently exploring the development of common best practices for nonclinical safety (testing and specification requirements) and creating a process to draft a well-defined, nonclinical data package for novel excipients [94]. USP also supports a novel excipient review program, which contributes to establishing new pathways for the development, and facilitating innovation for the advancement of, new medical products [95,96].

## 3. Compare and Contrast Country-Specific Perspectives

Excipient regulation, disclosure statements of excipient in pharmaceutical products, and other measures for safe excipient use in each country and region are summarized in Table 1. The regulatory processes for excipients included in the pharmaceutical product are similar between jurisdictions; however, information gaps remain. There are some resources for the judgment of excipient use in pediatric formulation through the regulatory process in the US or EU. However, there is no available database elsewhere. EMA and FDA proactively published several guides about acceptable daily intake of some excipients; however, no guidance or guidelines exists in other regions. The law and process for pediatric drug development may make a difference among the countries and regions.

The disclosure status of information on pharmaceutical excipients was similar among countries; that is, not all quantitative data was disclosed, and some excipients, such as pH adjusters and flavoring agents, were not specified in the product information. Some coloring agents and sweeteners that are not approved in one country were used in other countries. The function of sugar in formulation and amount of alcohol used are recommended to be stated in the package insert in the European countries and South-Africa, but there is no regulation for it in the other regions. As with any other special measures for the safety of the use of excipients for pediatric patients, the US and EU are attempting to create a database that contributes to the safety assessment of the use of excipients in the pediatric population.

Regarding the ‘threshold’, MDI, or contraindication of the use of excipients in the pediatric population, most countries declined to specify because the evidence-based quantitative information on the use of excipients in the pediatric population is insufficient.

## 4. Past and Ongoing Collaborative Efforts

Summary of attempts or ongoing efforts in each country and region was indicated in Table 2.

### 4.1. Workshop for the Safety Qualification of Excipients

The sessions were held on the 8th and 9th of June 2016 at a public workshop titled “Challenges and Strategies to Facilitate Formulation Development of Pediatric Drug Products.” [44]. The existing state and gaps, as well as ideas for risk-based techniques to facilitate the development of pediatric age-appropriate pharmacological products, were discussed during this session. The goal of the session was to bring together a diverse group of stakeholders (e.g., EU and US-based formulators, regulators, clinicians, and toxicologists) to discuss approaches to excipient safety assessment and to identify gaps and challenges in current paradigms for assessing excipient safety and evaluating potential risk in pediatric formulations. The necessity of a systematic, risk-based, proportional approach to safety evaluations was underlined by the participants throughout this event. The proposed risk-based strategy should only be utilized when an excipient is expected to be critical to the formulation’s performance. The interpretation of all the data might lead to recommended measures for excipients with a high toxicity potential for children, more research to better understand the potential dangers, or clinical monitoring of exposure or biomarkers of safety. Using orthogonal data sources, collaborative data sharing, and better awareness of existing sources, such as the STEP database and IID, were all considered significant in this session to close the gap in excipient information needed for risk assessment. The workshop’s organizers and attendees emphasized the need of establishing risk-based approaches for excipient safety evaluations, as well as the importance of meaningful stakeholder (e.g., patient, caregiver) involvement in pediatric formulation development.

### 4.2. The Safe Excipient Exposure in Neonates and Small Children (SEEN) Project

The Safe Excipient Exposure in Neonates and Small Children (SEEN) project was a retrospective cohort study. Based on a chart audit of multi-medicated patients under the age of 5, the SEEN project quantifies the total amount of excipients administered to poly-medicated neonatal and pediatric patients during hospitalization, and investigate whether any medical diseases are treated in European countries with potentially harmful excipients. As part of this project, the cumulative daily alcohol exposure (mg/kg/day) in polymedicated neonates and infants was measured. [97]. The findings revealed a lack of understanding of the acceptability of various dose forms, tastes, and, more crucially, the safety of formulation excipients in relation to children’s age and developmental stage.

### 4.3. The European Study of Neonatal Excipient Exposure (ESNEE)

At the end of 2009, a group of neonatal and pharmaceutical professionals from around Europe (Liverpool, Leicester, Belfast (UK); Paris (France); Tartu (Estonia)) gathered to discuss the present state and challenges of newborn excipient exposure. [98]. Their major goal was to give evidence for discussion about excipient. The consortium was formed, and the European Study of Neonatal Exposure to Excipients (ESNEE) was launched, by supporting from “Priority Medicines for Children (PRIMEDCHILD)”. ESNEE is a research project aimed at developing a set of procedures that will allow for an integrated assessment of neonatal exposure to potentially hazardous excipients in pharmaceutical products in Europe. The project creates new methodologies and gives knowledge that is needed for formula development and application. The ESNEE program comprises the following six work packages: (1) to undertake a comprehensive, European-based questionnaire and a point prevalence survey, of excipient exposure in neonates, to highlight opportunities for product substitution and priorities for reformulation; (2) to conduct a systematic review to identify existing information about the impact of excipients on the development of human neonates and juveniles in other species; (3) to develop techniques that allow small-volume blood samples to be used in population excipient kinetic (EK) models for systemic excipient exposure in human neonates; (4) to conduct a cohort study of neonates exposed to selected excipients, including blood samples for EK assays; (5) to develop EK models for selected excipients; (6) to integrate the results of the objectives, in work packages one to five, to detect formulation problems associated with the use of excipients in neonates Through this ESNEE program, Nellis et al. surveyed excipient uses in Europe in 2015 but found that manufacturers were reluctant to share the quantitative information for many products [99]. Additionally, Mulla et al. conducted a kinetic study for an excipient in the target population, for methyl and propyl parabens, in the same year [100].

### 4.4. Paediatric Excipient Risk Assessment (PERA) Framework

Selecting excipients with appropriate safety and tolerability is a major hurdle in pediatric formulation development. The suitability of a particular excipient will be dependent on the context of its users, such as the pediatric age range, acute versus chronic use, clinical risk/benefit of the disease, activity level, and excipient type. Scientists are encouraged to apply the principle of risk/benefit balance to assess the suitability of excipients to the specific pediatric population. An indicative list of parameters that should be taken into consideration and a hierarchy of information sources, when assessing the excipients’ risks, are provided by regulatory agencies. However, the approach to be taken and details of how the risk evaluation should be undertaken are lacking. Recently, a systematic approach called the “Paediatric Excipient Risk Assessment (PERA) framework”, to guide the selection of excipients and assessment of the risk of excipient exposure, has been developed through collaboration between IQ Consortium and EuPFI pediatric excipient sub-groups, which will be published soon [44]. The application of the PERA framework is key to both efficient product development and regulatory decision making. Proper application of PERA framework can lead to better communication and optimal discussions between excipient manufacturers, pharmaceutical product manufacturers, and regulatory agencies.

### 4.5. Harmonization of Pharmacopoeia 

The harmonization of standards among the pharmacopeia is one way to reduce this burden. In 1989, the Pharmacopeial Discussion Group (PDG) was formed, with representatives from the organizations that developed the JP, Ph. Eur., and USP–NF, to work toward the harmonization of pharmacopeial standards, such as excipient monographs [101,102]. Harmonization status can be referred to on the website of the Council of Europe. The published table summarizes the sign-off coversheets for all monographs of excipients under the PDG work plan. These coversheets provide detailed, helpful information about harmonized parts and local requirements for all individual texts, having undergone harmonization by the PDG.

### 4.6. Other Efforts

Several authorities attempt to refer to the guidelines or guidance published in the other countries or regions for the safety assessment of excipients. Harmonizing the ICH guidelines and the use of the open-source database may accelerate the excipient regulation process. Not all countries and regions introduce this system, and further improvement is needed. 

Regarding the ICH guidelines, the ICH S11 recommends an approach for the nonclinical safety evaluation of pharmaceutical excipients intended for development in pediatric populations [59]. To assess the safety of the excipients in a pediatric clinical formulation, available information on the excipients should be evaluated, and a weight of evidence approach (an assessment based on the entirety of the evidence, including the pharmacology, pharmacokinetic (ADME), nonclinical in vitro and in vivo animal studies, and the safety data from clinical settings) should be followed. If sufficient data to support the use of the excipient in the intended pediatric population is not available, further safety evaluation can be required. e.g., evaluating the excipient alone in a juvenile animal toxicity study. Since these guidelines only focus on nonclinical evaluations of pharmaceuticals including excipients’ safety for pediatrics, implementing the ICH guidelines may not always ensure excipients’ safety in clinical settings.

A real-world excipient exposure in preterm infants and neonates has previously been assessed, for several substances in various regions, by referencing product information [74,84,85,88,98,99,100,101,102,103,104,105,106,107,108,109,110,111,112,113,114,115]. However, limited information makes it impossible to evaluate the actual risk of excipients to the pediatric population.

Attempts of quantitative evaluations of excipient exposure, by using the donated blood samples, have also been tried [116,117,118] to evaluate an accurate and actual influence of excipients on pediatric patients, including neonates.

## 5. Discussion

In this review, we compared country-specific perspectives, including the current state on the safety assessment of pharmaceutical excipients, in formulations for both adults and pediatrics (including the disclosure status of excipients in pharmaceutical products,) and challenges in excipient regulation. Additionally, ongoing efforts for ensuring the safety of excipients for the pediatric population were summarized, and further possibilities of collaboration worldwide were discussed.

Manufacturers’ SmPC’s, package inserts, and PILs may be useful for identifying the excipients in particular medicine, and they may help determine the specific amounts of excipients present in pediatric formulations. However, there are many excipients in approved medicines that contain undeclared additives and concomitant components because excipient manufacturers have not been willing to disclose the identity of such components, due to the proprietary nature of their use, and there is no obligation of indication. Not all quantitative data is available in all countries or regions, and proper risk assessment has not been utilized for safety assessments for pediatric patients. Although some excipients were disclosed with quantitative information, and the extents of exposure were evaluated through dedicated investigations, the criteria and evaluation results were ambiguous. Furthermore, because most prescriptions for neonates, including preterm neonates, are “off-label” [14,45,113,121], there was no stringent regulation for manufacturers to identify the safety of the excipients in this pediatric population. As shown in the previous review report, toxicities and adverse effects of major used pharmaceutical excipients, on pediatric patients, were summarized [14]. Safety assessment is difficult and complicated. Safety concerns on the use of excipients are dependent on each patient’s background, such as age, including postmenstrual weeks and underlying disease (i.e., sweeteners have a risk for diabetes [122], saccharin is recommended to use only for children greater than 3 years [42]). Furthermore, the severity of toxicity caused by the excipients’ exposure is different. A risk assessment should be done, through the drug development and regulatory process, before use in clinical settings. Juvenile toxicity study can also be required, extensively, to assess the toxicities or sensitivities of excipients to pediatrics, even when the drugs used for pediatrics are expected [14]. The current status of challenges on excipient safety for pediatrics, and its solution, was summarized in Table 3.

In the regulatory process, the excipients included in the pharmaceutical products are reviewed by regulatory authorities in each country or region. However, background information on excipient safety for the pediatric population is lacking. As shown in the attempts of the EU, guidelines on excipient use and its labeling in the package leaflet of medicinal products will be needed for each region. More preferably, preparing the common and harmonized guidelines, or guidance, for excipient use and its labeling in the package leaflet will be desired. As shown in the recent review [66], various criteria were set based on the several guidelines for each excipient. The pharmacopoeias among Japan, EU, and the US, which are the base of the excipients’ monograph in each country, have not been harmonized [123]. The evidence base and determination process for the recommended doses is also different. Common harmonized guidelines and a unified excipients database may be helpful for regulatory authorities and healthcare professionals that are dedicated to pediatric patients. In addition to the safety profile, a list of excipients that can or cannot be used for pediatric pharmaceutical products will be helpful. The stakeholders in many countries are confronting common problems. Sharing those issues and hammering out effective measures would be beneficial. Looking into the current situation all over the world, through this review, may help the stakeholders overcome the current situation.

## 6. Future Perspectives of Pharmaceutical Excipients in Pediatrics Authors

To resolve this situation, a survey based on real-world prescription data and a quantitative risk assessment, by academia and clinical healthcare professionals, will be needed. As described in this review, information availability varies among countries in the world, and quantitative information on excipients, and their safety for pediatric patients, are rarely specified. As shown in several open databases (i.e., STEP database, IID), the enhancement of accessibility for data on excipients was found; however, evidence-based quantitative data for tolerated daily intake of each excipient for the pediatric population is still lacking. Clearer safety limits and quantitative information, for the problematic excipients in the pediatric population, are needed to aid healthcare professionals in drug selection for these patients. This is especially important in neonates and young children, as well as when patients are taking multiple, and long-term, medications, when considering the potential cumulative adverse effects.

For the evidence-based excipient regulation, collecting the evidence-based data for the safety of excipients in the pediatric population, gathering information on currently used excipients in pharmaceutical products, including quantitative information, and sharing the current issues of excipient exposure in pediatric patients with all stakeholders, including regulatory authorities in every country or region, is imperative. Additionally, a harmonized guideline with clearer safety limits and quantitative information, on excipients of concern, in the pediatric population for each country or region, will be needed. Internationally harmonized excipients’ regulatory processes may contribute to ensuring safe medicinal treatment for the pediatric population.

## 7. Limitations

As the limitation of this study, a limited number of regions was selected. For example, in Africa, we only reviewed the current situation on pharmaceutical excipients’ regulation in 2 out of 54 African countries. The same applies to the other regions. Further investigation for understanding the current situation all over the world will be required.

## Figures and Tables

**Figure 1 children-09-00453-f001:**
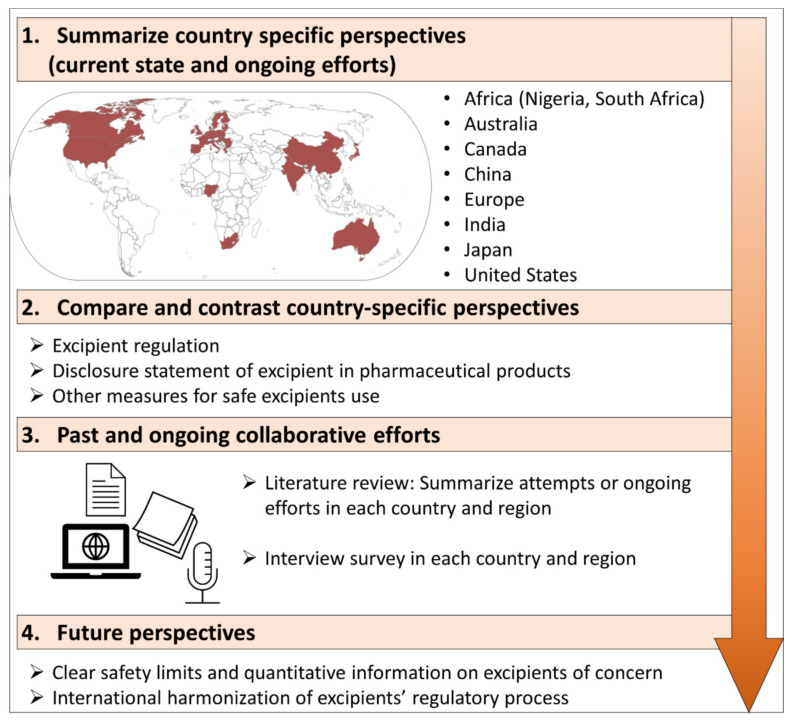
Concept and summary of this study.

**Table 1 children-09-00453-t001:** Excipient regulation, disclosure statements of excipient in pharmaceutical products, and other measures for safe excipient use in each country and region.

Country/Region	Regulation on Pharmaceutical Excipients	Disclosure of Excipients Information	Quantitative Information	Any other Special Measures for Safety of Excipients Use for Pediatrics
Africa	Nigeria: Reviewed by NAFDAC based on several guidelines and regulations in other authorities South Africa Refers to the EC guidelines on excipients use	Listing the composition of excipients in the package insert is not specified in the guidelines	Not for all excipients.	None
Australia	Reviewed by TGA following the EC guidelines	Only excipients specified in TGO No. 79 are required. Reference to color, fragrance, or flavor is generally considered to be acceptable without justification	Not for all excipients.	ARTG Excipients Project that allows all information on excipients available from a single source
Canada	Reviewed by Health Canada following the Canadian F&D Act and the F&D Regulations	The full qualitative composition The list of excipients which must feature on the labelling of medicinal products and the way in which these excipients must be indicated Some concerned excipients are described in the excipient warnings section and other sections in the package leaflet	Not for all excipients.	As per ICH guidance
China	Reviewed based on the Drug Administration Law of the People’s Republic of China	All excipients shall be listed for injections and non-prescription drugs The excipients included in a prescription, which may cause severe adverse reaction, shall be specified	No specific regulation.	
EU	Reviewed by EMA following the EC guidelines	The full qualitative composition The list of excipients which must feature on the labelling of medicinal products and the way in which these excipients must be indicated Some concerned excipients are described in the excipient warnings section from other sections in the package leaflet. ‘Threshold’ values are indicated in the guidelines	Not for all excipients.	STEP database
India	Reviewed based on the Cosmetics Act 1940 and Rules 1945 IPEC India support the excipients regulation in India	Partial information is available in the package insert leaflet	No specific regulation	Toxicological data from IPEC India are referred to the regulation process.
Japan	Reviewed by MHLW based on JP and JPE	All excipients included in the product must be listed Not mandatory to disclose the contents of flavoring and pH adjusters Excipients have maintained the confidentiality of proprietary information	Available for only parenteral formulation Not for all excipients	None
US	Reviewed by FDA following the FDA guidance or a novel excipient review program Flavoring agents are evaluated by FEMA independently	Excipients that may cause hypersensitivity or other adverse reactions need to be included along with the amount Excipients with a significant safety concern in pediatric patients must be described in package insert	Not required for all excipients.	FDA IID

MHLW, Ministry of Health, Labor, and Welfare; JP, Japanese Pharmacopoeia; JPE, Japanese Pharmaceutical Excipients; US, United States; FDA, US Food and Drug Administration; FEMA, Flavor and Extract Manufacturers Association; IID, Inactive Ingredient Database; EU, European union; EMA, European Medicines Agency; EC, European Commission; STEP, Safety and Toxicity of Excipients for Paediatrics; IPEC, International Pharmaceutical Excipients Council; F&D, Food and Drug; ICH, International Conference on Harmonization; NAFDAC, National Agency for Food and Drug Administration; TGA, Therapeutic Goods Administration; TGO, Therapeutic Good Order; ARTG, Australian Register of Therapeutic Goods.

**Table 2 children-09-00453-t002:** Summary of attempts or ongoing efforts in each country and region.

Country/Region	Other Attempts or Ongoing Efforts
Africa	Study on excipients exposure in pediatrics [23]
Australia	None
Canada	None
China	None
EU	Study on excipient exposure in pediatrics [67,97,99,111,112,115,119] ESNEE project [120] SEEN project [97] Workshop for the Safety Qualification of Excipients [44]
India	Study on excipients exposure in pediatrics [74]
Japan	A nationwide study on excipients exposure in neonates [85]
US	Study on excipients exposure in pediatrics [100,105] Incude pediatric safety in FDA IID Nove excipient review pilot program by CDER Workshops, survey, and pediatric excipient risk assessment framework development by IQ pediatric consortium

US, United States; FDA, US Food and Drug Administration; IID, Inactive Ingredient Database; EU, European union; ESNEE, European Study of Neonatal Exposure to Excipients; SEEN, Safe Excipient Exposure in Neonates and small children.

**Table 3 children-09-00453-t003:** Safety issues on pharmaceutical excipients for pediatric patients.

Challenges	Lack of evidence based safety data considering physiological, toxicokinetic, and toxicodynamic changes in pediatrics.
	Lack of evidence based safety data for the special population (i.e., preterm neonates, patients with specific disease).
	A safety evaluation of excipients in not only a pediatric formulation but also off-label used products is necessary before use referring to accessible safety data.
	Because accessible data are from adult human and animals, safety data from pediatric use and juvenile toxicity studies will be required.
Solution	The evidence based safety information of excipients should be included into the repository database as an accessible information on stakeholders

## Data Availability

Not applicable.

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
