# Peer review of "The Current States, Challenges, Ongoing Efforts, and Future Perspectives of Pharmaceutical Excipients in Pediatric Patients in Each Country and Region"

_children, 2022, doi:10.3390/children9040453_

Round 1

Reviewer 1 Report

1.      Page 2: Safety and biopharmaceutical challenges of excipients in pediatric formulations have been concerned for a long time (please check grammar)

2.      Page 2: Many excipients have no available safety data available to justify their use during regulatory approval of the pediatric drug products. (Please check use of available twice in the sentence)

3.      Page 2: Furthermore, the guidance or recommendation on excipi-87 ents use for the pediatric population vary by countries around the world (Please end sentence with a full stop/period)

4.      Page 3: The GRPP, a general guideline for all pharmaceutical products, includes a consideration of excipients for children in referring to…. (please check “in”, it appears out of place)

5.      Page 4: It 166 requires pH adjusters to be listed, even if not these are not present in the final product. (Sentence needs reworking)

6.      Page 4: Australia: Therapeutic Goods Administration (TGA) in the Australian Government Department of Health Australian (Please delete agency name to make headings uniform with other sections)

7.      Page 5: Canada: Health Canada (Please delete agency name to make headings uniform with other sections)

8.      Page 14: Please US in full i.e., United States of America

Author Response

February 24, 2022

Mr. Hugh Hu

Assistant Editor

Children

RE: children-1613579

Dear Mr. Hu:

The manuscript entitled “The current states, challenges, ongoing efforts, and future perspectives of pharmaceutical excipients in pediatric patients in each country and region” has been revised according to the Editor’s and Reviewers’ suggestions. Point-by-point responses to each of the queries have been provided below. Please note that for this letter, we have paraphrased each of the Reviewers’ questions and suggestions.

For the detail of the revision, please see the attached document.

We would like to thank the Editor and Reviewers for their insightful suggestions and comments. We hope that the modifications made to this manuscript will now make it acceptable for publication in Children.

Thank you very much once again for your consideration.

Sincerely,

Jumpei Saito, PhD

Department of Pharmacy

National Center for Child Health and Development

2-10-1 Okura, Setagaya-ku, Tokyo 157-8535, Japan

Tel.: +81-3-3416-0181 Fax: +81-3-3416-2222 E-mail: Saito-jn@ncchd.go.jp

Reviewer 2 Report

Reviewer’s Comments

The review paper can have a potential interest to pharmaceutical scientists working in the development of pediatric formulations, regulatory bodies to harmonize the use of excipients in pediatric formulation. It contains detail information including comparative study among different countries. However, the text lacks in-depth scientific discussion which need to be extended. Authors cannot focused only on the simple reporting of the research papers findings but also need to analyze their findings. I suggest critically working through the text with this question in mind and doing necessary changes

  1. Line 81-82 is already included in Line 27-28 “A major hurdle in pediatric formulation development is the lack of safety and toxicity data on some of the commonly used excipients.” Please delete here or in the introduction to avoid repetitions.
  2. Even though the topic includes the future perspectives of pharmaceutical excipients in pediatrics authors did not clearly cover this portion in the text. I suggest authors to add more information and discussion on this section. Also, the authors can also add their suggestion and recommendation.
  3. Authors can add and discussion more on the excipients that can have safety issues in pediatric patients ( In tabular form)
  4. Figure, schematic diagram, or flow chart are not included in the manuscript which make the paper more presentable and easy to understand the facts and figures compared to the reading the text and are less time consuming. Therefore, I recommend to add these things wherever it is possible in the manuscript.

Author Response

(The authors gave the same response as above.)

Round 2

Reviewer 2 Report

Thank you for considering each of the corrections and recommendations that I provided.